# mRNA Splicing of *UL44* and Secretion of *Alphaherpesvirinae* Glycoprotein C (gC) Is Conserved among the *Mardiviruses*

**DOI:** 10.3390/v16050782

**Published:** 2024-05-15

**Authors:** Huai Xu, Widaliz Vega-Rodriguez, Valeria Campos, Keith W. Jarosinski

**Affiliations:** Department of Pathobiology, College of Veterinary Medicine, University of Illinois at Urbana-Champaign, Urbana, IL 61802, USA; huaixu2@illinois.edu (H.X.); widaliz.v@gmail.com (W.V.-R.); valecam3@gmail.com (V.C.)

**Keywords:** herpesvirus, glycoprotein C, mRNA splicing, Marek’s disease (MD), vaccines

## Abstract

Marek’s disease (MD), caused by *gallid alphaherpesvirus 2* (GaAHV2) or Marek’s disease herpesvirus (MDV), is a devastating disease in chickens characterized by the development of lymphomas throughout the body. Vaccine strains used against MD include *gallid alphaherpesvirus* 3 (GaAHV3), a non-oncogenic chicken alphaherpesvirus homologous to MDV, and homologous meleagrid alphaherpesvirus 1 (MeAHV1) or turkey herpesvirus (HVT). Previous work has shown most of the MDV gC produced during in vitro passage is secreted into the media of infected cells although the predicted protein contains a transmembrane domain. We formerly identified two alternatively spliced gC mRNAs that are secreted during MDV replication in vitro, termed gC104 and gC145 based on the size of the intron removed for each *UL44* (gC) transcript. Since gC is conserved within the *Alphaherpesvirinae* subfamily, we hypothesized GaAHV3 (strain 301B/1) and HVT also secrete gC due to mRNA splicing. To address this, we collected media from 301B/1- and HVT-infected cell cultures and used Western blot analyses and determined that both 301B/1 and HVT produced secreted gC. Next, we extracted RNAs from 301B/1- and HVT-infected cell cultures and chicken feather follicle epithelial (FFE) skin cells. RT-PCR analyses confirmed one splicing variant for 301B/1 gC (gC104) and two variants for HVT gC (gC104 and gC145). Interestingly, the splicing between all three viruses was remarkably conserved. Further analysis of predicted and validated mRNA splicing donor, branch point (BP), and acceptor sites suggested single nucleotide polymorphisms (SNPs) within the 301B/1 *UL44* transcript sequence resulted in no gC145 being produced. However, modification of the 301B/1 gC145 donor, BP, and acceptor sites to the MDV *UL44* sequences did not result in gC145 mRNA splice variant, suggesting mRNA splicing is more complex than originally hypothesized. In all, our results show that mRNA splicing of avian herpesviruses is conserved and this information may be important in developing the next generation of MD vaccines or therapies to block transmission.

## 1. Introduction

Marek’s disease (MD) is a chicken disease that can be devastating to chickens. It is characterized by severe clinical signs that include immune neurological disorders, the development of lymphomas predominantly in the viscera and muscles by transforming T cells, and immune suppression. MD is caused by gallid alphaherpesvirus (GaAHV) 2 (GaAHV2), better known as Marek’s disease virus (MDV), a member of the subfamily *Alphaherpesvirinae*, species *Mardivirus gallidalpha2* and the prototypic member of the *Mardivirus* genus [1].

MDV infection has four phases: primary cytolytic replication which is followed by the establishment of latent infection, reactivation, and ensuing virus replication or secondary infection [2]. MDV infection starts from the inhalation of infectious virus through the respiratory route by the chicken. B lymphocytes and macrophages [3] are the primary target cells for infection which transports the virus to lymphoid organs and skin after infection. Depending on the line of chicken and the virulence of the virus, a small proportion of latently infected T cells can be transformed, resulting in tumor formation, and ultimately the death of the chicken. It is crucial for the MDV life cycle that infected immune cells circulating to the periphery transfer the virus to specialized skin cells called feather follicle epithelial (FFE) cells, as these cells are where MDV is shed from the chicken into the environment in the form of infectious dander. This completes the virus life cycle that facilitates constant spread to naïve chickens.

MD is controlled by vaccination with homologous non-oncogenic avian herpesviruses or attenuated MDV strains in chickens. Gallid alphaherpesvirus 3 (GaAHV3), species *Mardivirus gallidalpha3*, and meleagrid alphaherpesvirus 1 (MeAHV1), species *Mardivirus meleagridalpha 1* or turkey herpesvirus (HVT) are widely used viruses that have similar life cycles as MDV. Although effective at reducing disease incidence, current vaccines do not block the spread of MDV in poultry houses. This has resulted in increasing virulence over the decades [4].

*Alphaherpesvirinae* conserved glycoprotein C (gC) are type 1 membrane proteins and not essential for cell culture propagation, but they play an important part in the initial attachment of cell-free viruses to heparin- and chondroitin sulfate proteoglycans on the surface of cells [5,6,7]. Another function is their involvement in the final stages of virus egress from cultured cells [6,8]. The gC proteins of herpes simplex virus (HSV) 1 (HSV-1) and 2 (HSV-2), suid alphaherpesvirus 1 (SuAHV1) or pseudorabies virus (PRV), equid alphaherpesvirus 1 (EqAHV-1), and bovine alphaherpesvirus 1 (BoAHV-1) are thought to have immune evasion functions through binding to, and inhibiting, the action of complement component C3 [9,10,11,12,13], independent of their role in virus attachment and egress. Although gC is not essential for in vitro propagation, it is essential for horizontal transmission or natural infection of MDV and GaAHV3 (301B/1 strain) in chickens [14,15] and HSV-1 and VZV replication in human skin cells [16].

Approximately 95% of the MDV gC protein expressed in cell culture is secreted in the medium of infected cells [17], despite the predicted full-length MDV gC encoding a transmembrane domain, consistent with other gC homologs. Two alternative mRNA splice variants termed gC104 and gC145 were identified by reverse transcription (RT)-PCR analyses and confirmed to produce secreted forms of gC in cell culture due to 104 and 145 nt introns and premature stop codons [18]. Importantly, when each form of gC was expressed individually, all three (gCfull, gC104, and gC145) were required for efficient horizontal transmission of MDV, suggesting each protein plays an important role in transmission. Recently, it was confirmed all three gC transcripts are produced in FFE skin cells with high abundance and unique peptides were identified for gC104 confirming its expression in vivo [19].

The alternative mRNA splicing of *UL44* (gC) is not specific to MDV. A secreted form of HSV-1 gC is produced by the alternative mRNA splicing of *UL44* with a portion of the *UL45* [20]. The lack of the ICP27 immediate early protein was necessary for this splicing [21]. Recently, we and others showed that the expression and splicing of MDV gC are regulated by ICP27 and UL47 [22,23].

Here, we asked whether 301B/1 and HVT produce secreted gC, similar to MDV. Using epitope-tagged gC, we detected secreted forms of 301B/1 and HVT gC in cell culture. RT-PCR analyses confirmed that both viruses produce mRNA splice variants in cell culture and FFE skin cells with both viruses producing gC104, while only HVT produced gC145. Interestingly, the donor sites for gC145 are 66% conserved between MDV, 301B/1, and HVT, while gC104 is 100% conserved. We hypothesized a single nt difference (G > C) in the donor sequence of 301B/1 gC145 would explain why no gC145 is produced. However, after modifying the donor site (C > G) in 301B/1, gC145 is still not produced. Next, we hypothesized the lack of gC145 transcript was due to a single nt difference (C > T) in the *UL44* branch point (BP). However, modification of this sequence also did not result in the production of gC145 transcripts. To be complete, we replaced the gC104 and gC145 common acceptor site of 301B/1 with that of MDV. Still, the gC145 transcript was not produced, despite replacing the donor, BP, and acceptor site with the MDV gC sequences. Recently, ICP27 and UL47 have been implicated in regulating the mRNA splicing of MDV gC [22,23]. Our results suggest that mRNA splicing of *UL44* by *Mardiviruses* is likely more complicated than originally thought and implicates virus-specific regulation through ICP27 or UL47.

## 2. Materials and Methods

### 2.1. Cell Cultures

Chicken embryo cell (CEC) cultures were prepared with 10–11 days old specific-pathogen-free (SPF) eggs using standard methods [24]. Primary CEC cultures were grown in a Medium 199 (Cellgro, Corning, NY, USA) growth medium supplemented with 10% tryptose-phosphate broth (TPB), 0.63% NaHCO_3_ solution, antibiotics (100 U/mL penicillin and 100 µg/mL streptomycin), and 4% fetal bovine serum (FBS), and then changed to 0.2% FBS when the cells were confluent.

The chicken DF-1-Cre fibroblast cell line [25] was seeded in a 1:1 mixture of Leibovitz L-15 and McCoy 5A (LM) media (Gibco, Gaithersburg, MD, USA) supplemented with 10% FBS plus antibiotics (100 U/mL penicillin and 100 µg/mL streptomycin) and 50 µg/mL Zeocin (Invitrogen, Carlsbad, CA, USA).

All cells were maintained in a humidified atmosphere of 5% CO_2_ at 38 °C.

### 2.2. Viruses

Recombinant (r) GaAHV3 301B/1 r301B47R, r3ΔgC, r3ΔgC-R, and r3-MDVgC have been previously described and characterized [15], while rHVT were generated in this report and described below. A BAC clone containing the genome of HVT strain FC126 was obtained from Elanco Animal Health (Greenfield, IN, USA). The BAC was made by insertion of a mini-F cassette between *UL44* and *UL45* using standard molecular biology techniques. The mini-F cassette was flanked by *lox*P sites and expressed Cre under a eukaryotic promoter such that the mini-F was excised upon transfection into CEC cultures.

### 2.3. Western Blot Analysis of Secreted gC Proteins

Western blot analyses were performed essentially as previously described for MDV gC [26]. Cell culture media and total protein were collected from infected CEC cultures. To detect 301B/1 or HVT gC proteins, mouse anti-Flag M2 (Sigma-Aldrich, St. Louis, MO, USA) or rabbit anti-HA (Cell Signaling, Danvers, MA, USA) antibodies were used, respectively. Secondary anti-mouse or -rabbit Ig peroxidase conjugates were obtained from GE Healthcare (Piscataway, NJ, USA). The SuperSignal West Pico Chemiluminescent Substrate kit (Thermo Fischer Scientific, Rockford, IL, USA) was used to generate a chemiluminescent signal according to the manufacturer’s instructions.

### 2.4. RNA Extraction and RT-PCR Analysis

Total RNAs were collected using the RNA STAT-60 from Tel-Test, Inc. (Friendship, TX, USA) according to the manufacturer’s instructions. Briefly, MDV-, 301B/1- and HVT-infected CEC cultures in 75-cm^2^ tissue culture flasks with ~80% cytopathic effect, and MDV-, 301B/1- and HVT-infected FFE cells were scraped in RNA STAT60. RNA was DNase treated using the Turbo DNA-*free* kit (Thermo Fisher Scientific) and then stored at −80 °C until it was used for RT-PCR assays.

RT was performed with 1 µg of RNA using the High-Capacity cDNA Reverse Transcription Kit (Thermo Fisher Scientific) and RT reactions were carried out according to the manufacturer’s instructions with random priming. The reaction mixture was incubated at 25 °C for 10 min, then 37 °C for 120 min, followed by 85 °C for 5 min. To amplify cDNA, 3 µL of the RT mixture was mixed with DreamTaq Green PCR Master Mix (Thermo Fisher Scientific). Primers used for the amplification of the *UL44* are shown in Table 1. Differential primers were used to separate the different mRNA splice variants.

All RT-PCR mixtures were electrophoresed through 0.8 or 2% agarose Tris-acetate-EDTA (TAE) gels, and results were recorded by using an Eagle EyeII Still Video system (Stratagene, La Jolla, CA, USA). The *UL44* PCR products were purified from the agarose gel by using the QIAquick gel extraction kit (Qiagen, Inc., Germantown, MD, USA) and cloned by using the TOPO TA cloning kit (Life Technologies, Carlsbad, CA, USA) according to the manufacturer’s instructions.

### 2.5. DNA Sequencing

After the cloning of the RT-PCR products, both strands of DNA were sequenced for each construct. Sanger sequencing was performed by the DNA Services Facility at the Roy J. Carver Biotechnology Center at the University of Illinois at Urbana-Champaign. SnapGene 6.0.7 software (from Insightful Science; available at https://www.snapgene.com/, 14 May 2024) was used to analyze sequencing results.

### 2.6. Generation of Recombinant (r)HVT

To create rHVT expressing fluorescent-tagged UL47, the coding sequence of the enhanced green fluorescent protein (eGFP) gene was inserted at the C-terminus of the HVT UL47 ORF using two-step Red-mediated mutagenesis [27]. Briefly, the *eGFP*-*I*-*Sce*I-*aphAI* cassette was amplified from pEP-eGFP-in using primers shown in Table 2 and used for mutagenesis in GS1783 *Escherichia coli* cells. Multiple integrated and resolved clones were screened by restriction fragment length polymorphism (RFLP) analysis, analytic PCR, and DNA sequencing using primers shown in Table 1.

To create rHΔgC, the coding sequence of HVT *UL44* (gC) was deleted from rHVT47G. Briefly, the I-*Sce*I-*aphAI* cassette from pEP-KanSII was amplified by PCR with Thermo Scientific Phusion Flash High-Fidelity PCR Master Mix (Thermo Fisher Scientific, Waltham, MA, USA) using primers shown in Table 2 and used for mutagenesis in GS1783 *E. coli* cells. Following the removal of *UL44* in the rHVT47G clone, HVT gC or MDV gC were inserted into rHΔgC using two-step Red recombination. Briefly, HVT gC or MDV gC were PCR amplified from pEP-HVTgC-in or pEP-MDVgC-in, respectively, using primers shown in Table 2 and used for mutagenesis as described above. Two-step Red recombination was used to insert an HA epitope to the N-terminus of HVT gC after the predicted signal peptide sequence (CAG/LP). RFLP analysis, analytical PCR, and DNA sequencing confirmed all clones were correct. Primers used for MDV gC have been previously published [18,28,29], while primers for sequencing HVT gC are listed in Table 1.

To produce pEP-HVTgC-in shuttle vector, HVT *UL44* was amplified by PCR using a set of primers encompassing the complete *UL44* gene, gel purified, and cloned into the pcDNA3.1 vector (Life Technologies, Carlsbad, CA, USA) linearized using *Hind*III and *Xba*I restriction enzymes (New England Biolabs, Ipswich, MA, USA) to generate pcHVTgC. Next, the *I*-*Sce*I-*aphAI* cassette was amplified from pEP-KanSII using primers shown in Table 2 and inserted into pcHVTgC using restriction enzyme cloning utilizing the unique *Bam*HI site in HVT *UL44* to make pEP-HVTgC-in. This construct was used to reinsert HVT *UL44* into rHΔgC. To add an epitope tag, an HA epitope was inserted at the N-terminus of HVT *UL44* after the predicted signal peptide sequence to create rHaHVTgC using two-step Red recombination using primers shown in Table 2. All clones at each step were confirmed by PCR and DNA sequencing. For the insertion of MDV gC (RB1B strain) into rHVT, a previously described pEP-MDVgC-in shuttle vector was used [18].

rHVTs were reconstituted by transfecting primary CEC cultures with purified BAC DNA plus Lipofectamine 2000 (Invitrogen) using the manufacturer’s instructions, then further propagated in CEC cultures until virus stocks could be stored. All viruses were used at ≤5 passages for in vitro and in vivo studies.

### 2.7. Generation of Recombinant 301B/1 with Mutations in the Donor, BP, and Acceptor Sites

MDV, HVT, and 301B/1 gC donor sites and acceptor sites were predicted using https://www.fruitfly.org/seq_tools/splice.html, 14 May 2024. Multiple mutations in 301B/1 *UL44* were generated using two-step Red-mediated mutagenesis in GS1783 *E. coli* cells. Briefly, the I-*SceI-aphAI* cassette from pEP-KanSII was amplified by PCR using Thermo Scientific Phusion Flash High-Fidelity PCR Master Mix with primers shown in Table 3. Clones were confirmed by RFLP analysis, analytical PCR, and DNA sequencing.

r301B/1 was reconstituted by transfecting DF-1-Cre cells with purified BAC DNA plus jetOPTIMUS^®^ (Polyplus, New York City, NY, USA) using the manufacturer’s instructions. Fresh primary CEC cultures were mixed with transfected DF-1-Cre cells until plaques formed, then further passed in CEC cultures until virus stocks could be stored. All reconstituted BAC viruses were used at <5 passages.

## 3. Results

### 3.1. Generation of rHVT

#### 3.1.1. Generation of rHVT47G

Since fusing fluorescent proteins to the C-terminus of alphaherpesvirus UL47 (VP13/14) allows the visualization of infected cells and does not affect replication in cell culture and in vivo for numerous herpesviruses [15,30,31,32,33,34], we first generated rHVT with eGFP at the C-terminus of the HVT UL47. RFLP analysis confirmed the integrity of the BAC clones as the predicted banding pattern was observed as described in the figure (Figure 1a,b). In addition, DNA sequencing was used to confirm that each clone was correct at the nucleotide level using primers specific for each gene (Table 1).

#### 3.1.2. Generation of rHVT Lacking gC or Expressing HA-Tagged HVT gC or MDV gC

To analyze HVT gC expression, we first generated a gC-null rHVT in which the complete ORF of gC (*UL44*) was deleted (Figure 1c). Next, a gC-rescuent virus was generated where the complete *UL44* ORF was placed back into the *UL44* locus. Since we do not have HVT gC-specific antibodies, we placed an HA epitope at the N-terminus of the HVT gC ORF to generate vHΔgC-R (Figure 1d). In addition, a rHVT in which MDV gC replaced HVT gC (vH-MDVgC) was generated (Figure 1e). RFLP analysis confirmed the integrity of the BAC clones as the predicted banding pattern was observed (Figure 1b–e). In addition, DNA sequencing confirmed that each clone was correct at the nucleotide level using primers specific for each gene (Table 1).

### 3.2. Secretion of HVT gC Protein

We formerly showed that 301B/1 produced secreted gC similarly to MDV [15]. Additionally, we showed 301B/1 expressing MDV gC secrets MDV gC suggesting gC secretion is conserved between these two viruses. Here, we examined HVT and MDV gC expression in HVT-infected cells. Figure 2 shows Western blotting of cellular protein lysate and media from 301B/1 (Figure 2a) and HVT (Figure 2b)-infected CEC cultures. Similar to 301B/1 and MDV, HVT also secreted HVT (vH-HΔgC-R) and MDV (vH-MDVgC) gC into the media. These results show secretion of gC is conserved among MDV, 301B/1, and HVT.

### 3.3. Identification of HVT and 301B/1 UL44 (gC) mRNA Splice Variants

The production of secreted MDV gC in cell culture was shown to be due to mRNA splicing of the *UL44* transcript removing 104 and 145 nt introns to produce gC104 and gC145, respectively [18]. Recently, the expression of MDV gC splice variants was confirmed in FFE cells of MDV-infected chickens [19]. To determine whether secreted 301B/1 and HVT gC were due to mRNA splicing, RT-PCR analyses were performed on total RNA collected from 301B/1- and HVT-infected CEC cultures and FFE cells from infected chickens with samples from formerly published studies [15,22] using primers overlapping the start and stop codons of their respective *UL44* (Table 1). There appeared to be two bands at ~1.4 kb and <1.4 kb for both viruses and in both tissues (Figure 3a). Similar results have been shown for MDV gC [18]. Each band was gel purified, cloned, and individual clones sequenced. The largest band was identified as the full-length transcript of HVT and 301B/1 gC, while two alternative splice variants for HVT and one alternative splice variant for 301B/1 were cloned (Figure 3b).

To better separate the individual transcripts in RT-PCR assays, primers flanking the *UL44* introns for each virus were used in differential RT-PCR assays (Figure 3c). The expected RT-PCR products are summarized in Figure 3d. It was confirmed that 301B/1 did not express the gC145 transcript, while both MDV and HVT expressed both gC104 and gC145 in both CEC cultures and FFE skin cells in vivo. These results show that mRNA splicing of the conserved *UL44* (gC) transcript is conserved among MDV, HVT, and 301B/1.

### 3.4. Conserved mRNA Splicing Donor and Acceptor Sequences

Our differential RT-PCR analysis confirmed both MDV and HVT express gC145, while 301B did not produce this spliced variant. Using the Splice Site Prediction by Neural Network program [35], MDV, HVT, and 301B/1 *UL44* genes had 8, 4, and 5 predicted donor (Table 4) and 10, 14, and 11 acceptor (Table 5) sites, respectively. Interestingly, all three donor sites for the gC104 variant are 100% conserved (Figure 4), while the HVT gC145 donor sequences are less conserved comparing MDV and 301B/1 than comparing MDV and 301B/1. For all viruses, gC104 and gC145 utilize the same acceptor sites that are >85% conserved between all three viruses and all are functional based on all three viruses utilizing this acceptor site for production of their respective gC104 transcripts.

### 3.5. Modification of 301B/1 Donor, BP, and Acceptor Sequences Does Not Rescue gC145 Splicing

It was interesting that 301B/1 did not produce the gC145 splice variant considering both MDV and HVT produced this transcript. When comparing the sequences, there was a single nucleotide difference between a potential gC145 donor site for 301B/1 (cccgacCgtatatca) compared to MDV (cccgacGgtatatca) and HVT (accgacGgtatatta). Since gC104 and gC145 splice variants utilize the same BP and acceptor sites and they are clearly functional in 301B/1 producing gC104, we reasoned the C at position 1002 of the 301B/1 *UL44* made this donor sequence non-functional. To test this hypothesis, we mutated position 1002 from C to G in r301B47R, creating the same sequence as the MDV gC145 donor sequence generating v3-*UL44*-C1002G (Figure 5a). Following reconstitution and propagation in CEC cultures, there was no change in mRNA splicing for 301B/1 *UL44* with only unspliced and gC104 produced using our differential RT-PCR assay (Figure 5b).

Next, we modified the predicted BP for 301B/1 *UL44* (cctaaT) since it also only had one SNP compared to MDV *UL44* BP (cctaaC) in both wildtype v301B47G and v3-*UL44*-C1002G backgrounds. Again, after reconstitution and propagation in CEC cultures, both viruses containing the MDV gC145 and BP did not produce gC145 (Figure 5c). Interestingly, when MDV *UL44* (gC) was expressed by 301B/1, gC145 transcripts were produced.

Finally, we replaced the 301B/1 gC104/145 acceptor site with the MDV gC104/145 (Macc) in the background of v3-44-C1002G, v3-44-T1122C, and v3-44-C1002G/T1122C, with the last replacing the donor, BP, and acceptor sites of 301B/1 to MDV. Still, only gC104 was produced in 301B/1-infected CEC cultures (Figure 5d).

## 4. Discussion

The exact role that conserved gC proteins play during transmission is unknown currently, although all three forms of gC are needed for the efficient transmission of MDV in chickens [18]. Here, we showed that two other *Mardiviruses*, GaAHV3 and HVT also produce secreted gC proteins (Figure 2). Remarkably, both viruses produced the same size introns for gC104, while only HVT produced gC145 (Figure 3). To address the mechanistic reason for 301B/1 unable to produce gC145 transcripts, the 301B/1 *UL44* splice donor, BP, and acceptor sites were mutated to MDV gC, of which both MDV gC104 and gC145 are produced by 301B/1 (Figure 3 and Figure 5). The mechanistic reason for the lack of 301B/1 gC145 is still unknown but suggests other regions within the *UL44* transcript may mediate specific mRNA splicing.

For HSV-1, regulation of *UL44* (gC) splicing is mediated by ICP27 such that the lack of the ICP27 immediate early protein was necessary for this splicing [20]. That is, HSV-1 gC is not spliced and the full-length protein (gC1) is largely produced during infection when ICP27 is normally present, while non-functional ICP27 mutants lead to splicing *UL44* (gC) mRNA to produce a secreted form of gC, which they refer to as SEgC. However, for other members of the *Orthoherpesviridae*, ICP27 may promote splicing, as has been seen for Kaposi’s sarcoma-associated gammaherpesvirus (KSHV) homolog, ORF57 [36]. Thus, mRNA splicing mediated by herpesvirus is more complicated than originally thought.

The mechanism by which ICP27 homologs promote or inhibit mRNA splicing is not completely understood. For HSV-1, a responsive element in the pre-mRNA *UL44* transcript was found but it was not shown whether ICP27 binds to this element [21]. In another study, ICP27 was shown to promote the formation of an alternatively spliced form of promyelocytic leukemia [37]. More recently, the splicing of MDV gC104 and gC145 was differentially mediated by ICP27 and UL47 [23]. It is interesting that when MDV gC was expressed in 301B/1 (v3-MDVgC), all three forms of MDV gC were produced, while only 301B/1 unspliced and gC104 were produced in wildtype 301B/1 (Figure 3) after mutation of the gC145 donor, BP, and acceptor sites to MDV *UL44* sequences. These results suggest other sequences within the *UL44* transcript may regulate the splicing of gC104 and gC145. Further studies are required but it is tempting to speculate MDV, 301B/1, and likely HVT *UL44* sequences encode specific intron-responsive elements like what has been shown for HSV-1 *UL44* [21]. The hypothetical sequences may be differentially mediated by ICP27 and UL47 which were recently shown for MDV gC [23] and may be virus-specific.

## 5. Conclusions

In conclusion, our data confirm that mRNA splicing of *UL44* (gC) is conserved among the *Mardivirus*. Future studies will be directed at determining the mechanistic significance of differential splicing and secretion of *UL44* (gC) homologs in vivo and the apparent species-specific regulation of the conserved *UL44* (gC) transcript mRNA splicing. Understanding the role these individual gC proteins play in transmission will help in generating MD vaccines that block horizontal transmission of MDV in the field.

## Figures and Tables

**Figure 1 viruses-16-00782-f001:**
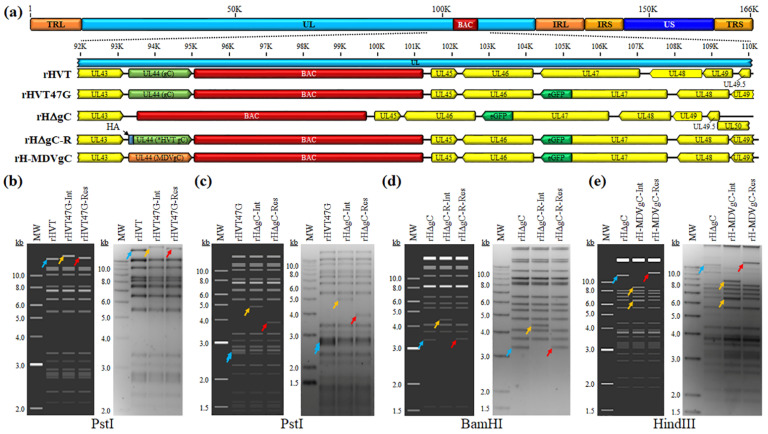
Generation of rHVT clones. (**a**) Schematic representation of the rHVT infectious clone genome with the terminal repeat long (TRL) and short (TRS), internal repeat long (IRL) and short (IRS), and unique long (UL) and short (US) regions. The region of the UL spanning *UL43* to *U49* is expanded to show the relevant genes within this region and modifications for each rHVT clone. (**b**–**e**) The predicted and actual RFLP analysis for generation of rHVT47G (**b**), rHΔgC (**c**), rHΔgC-R (**d**), and rH-MDVgC (**e**) clones. BAC DNA obtained from all clones was digested with *Pst*I (**b**,**c**), *Bam*HI (**d**), or *Hind*III (**e**) and electrophoresed through a 1.0% agarose gel. Integrates (Int) and resolved (Res) clones of the *I*-*Sce*I-*aphAI* sequence are shown. The predicted sizes are shown and are as follows: (**b**) Insertion of the *eGFP*-*I*-*Sce*I-*aphAI* sequence at the C-terminus of *UL47* in rHVT resulted in an increase in the 16,392 bp fragment (→) to 18,179 bp (→). The resolution of the *I*-*Sce*I-*aphAI* sequence reduced the 18,179 bp fragment by 948 bp to 17,195 bp (→) to create rHVT47G. (**c**) Insertion of the *I*-*Sce*I-*aphAI* sequence to remove the *UL44* ORF in rHVT47G resulted in the removal of a *Pst*I site combining 2693 and 2646 bp fragments (→) to create a single fragment of 4898 bp (→). Resolution of the *I*-*Sce*I-*aphAI* sequence reduced the 4898 bp fragment by 1026 bp to 3872 bp (→) to create rHΔgC. (**d**) Insertion of the *1* × *HaHVTgC* + *I*-*Sce*I-*aphAI* sequence in rHΔgC resulted in an increase in the 3334 bp fragment (→) to 4398 bp (→). Resolution of the *I*-*Sce*I-*aphAI* sequence reduced the 4398 bp fragment by 1028 bp to 3361 bp (→) creating rHΔgC-R. (**e**) Insertion of the *MDVgC* + *I*-*Sce*I-*aphAI* sequence into rHΔgC added a *Hind*III site resulting in shifting the 12,560 bp fragment (→) to 8687 and 6419 bp fragments (→). Resolution of the *I*-*Sce*I-*aphAI* sequence removed the additional *Hind*III site to create a single fragment of 14,072 bp (→) that is 2268 bp larger than the original 12,560 bp fragment creating rH-MDVgC. The predicted RFLPs were generated using SnapGene 6.0.7 software (from Insightful Science; available at snapgene.com). For actual RFLP analyses, the 1 kb Plus DNA Ladder from Invitrogen, Inc. (Carlsbad, CA, USA) was used as a molecular weight (MW) maker. No extraneous alterations are evident.

**Figure 2 viruses-16-00782-f002:**
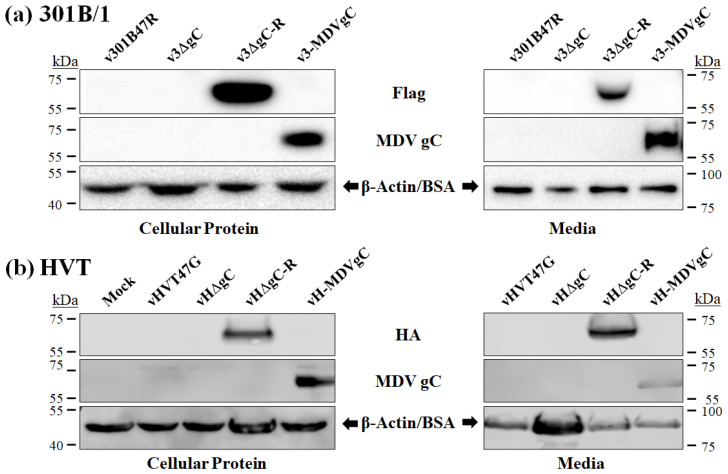
Both GaAHV3 301B/1 and HVT secrete their respective gC and MDV gC. Both total cellular protein and infected cell culture media were used to detect 3 × Flag or HA-tagged 301B/1 (**a**) and HVTgC (**b**), respectively, or MDV gC. Anti-Flag M2 and -HA were used to detect 301B/1 and HVT gC, respectively, while anti-MDV gC A6 antibody was used to confirm MDV gC expression. Previously described 301B/1 viruses include wild-type 301B/1 expressing UL47mRFp (v301B47R), v301B47R with *UL44* deleted (v3ΔgC), v3ΔgC rescuent with 301B/1 1 × Flag fusion gC protein (v2ΔgC-R) or MDV gC (v3-MDVgC). For protein loading control, mouse anti-β-actin was used for total protein. Anti-BSA antibody was used as a loading control for infected cell media.

**Figure 3 viruses-16-00782-f003:**
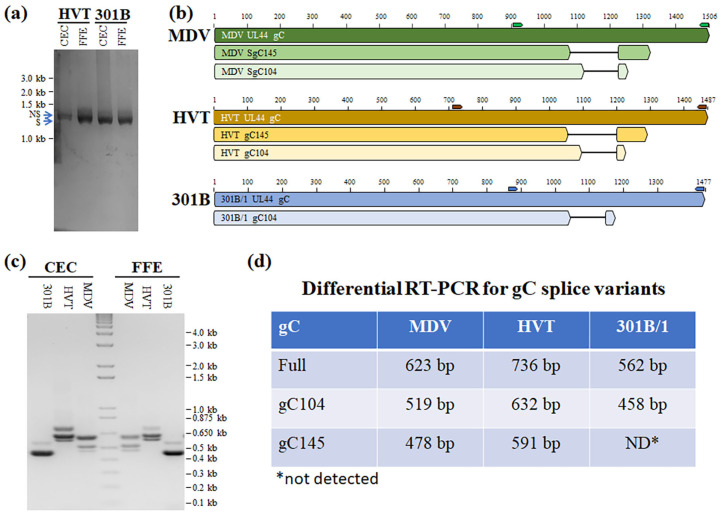
GaAHV3 (301B/1) and HVT produce *UL44* (gC) splice variants. Total RNA was collected from 301B/1 or HVT-infected CEC cultures or FFE cells and used in RT-PCR assays. Primers spanning the start and stop codons for 301B/1 and HVT *UL44* ORF (gC) were used in (**a**) showing non-spliced (NS) and spliced (S) products. RT-PCR products were cloned and sequenced. (**b**) HVT produces two splice variants with introns of 145 and 104 bp, while 301B/1 only expressed one variant with 104 bp intron. (**c**) Differential RT-PCR was used to better identify full-length gC, gC145, and gC104 mRNA splice variants using primers in Table 1 and 2% agarose gels. Primer locations are noted for each gene. (**d**) Summary of RT-PCR products identified during MDV, 301B/1, and HVT replication.

**Figure 4 viruses-16-00782-f004:**
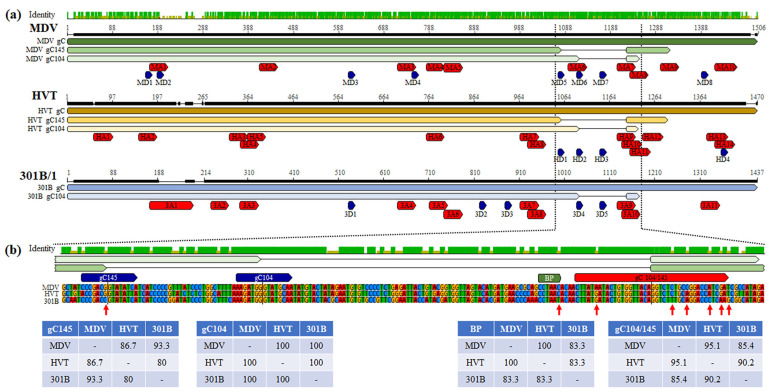
Conservation of *UL44* (gC) mRNA splicing donor and acceptor sites for MDV, HVT, and 301B/1. (**a**) Alignment of MDV (RB-1B), HVT (FC126), and GaAHV3 (301B/1) *UL44* gene sequences using MUSCLE Alignment in Geneious Prime 2021.0.3 (Biomatters, Inc., San Diego, CA, USA). The locations of donor (D) and acceptor (A) sites listed in Table 4 and Table 5 are shown, along with gC104 and gC145 splice variants identified. (**b**) The alignments spanning the gC145 donor, BP, and gC104/145 acceptor sites are expanded. SNPs targeted for modification are identified using a red arrow (→). Tables show the percent identity of the donor, branch point (BP), and acceptor sequences between MDV, HVT, and 301B/1.

**Figure 5 viruses-16-00782-f005:**
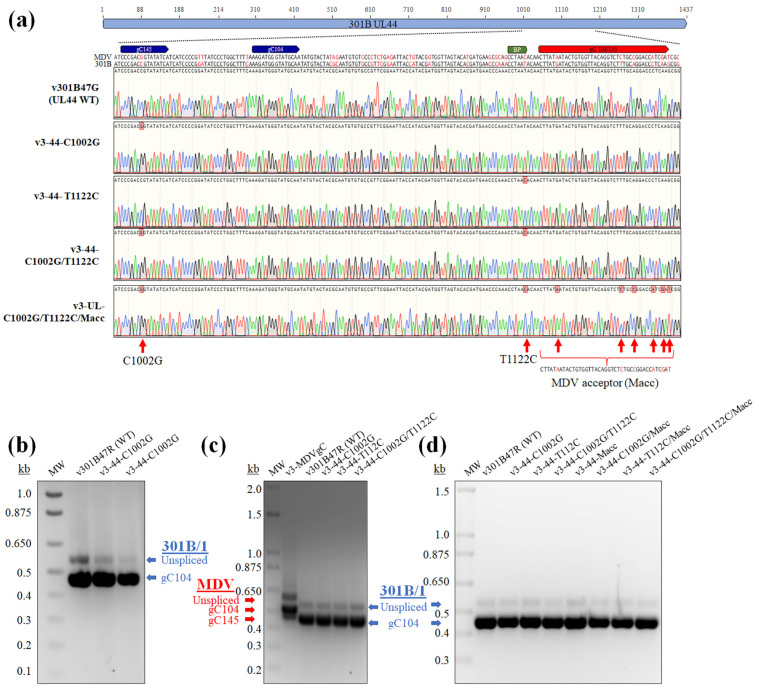
Modification of GaAHV3 301B/1 *UL44* gC145 donor, BP, and acceptor sites to MDV. (**a**) Region of 301B/1 *UL44* transcript expanded to show the region spanning gC splice variant motifs. Alignment of MDV (RB-1B) and GaAHV3 (301B/1) with differences in Red. Multiple mutations were generated to create MDV sequences in the 301B/1 gene including gC145 (C1002G), BP (T1122C), and multiple changes in the acceptor site (Macc). Mutations were incorporated into 301B/1 *UL44* and the virus was reconstituted (**b**–**d**). (**b**) Differential RT-PCR assays following modification of the gC145 donor site showed only unspliced and gC104 transcripts. (**c**) Differential RT-PCR assays following modification of the BP site (T1122C) in wildtype (v301B47R) and v3-44-C1002G backgrounds. v3 -MDVgC was included to show unspliced, gC104, and gC145 transcripts produced in 301B/1. (**d**) Summary of all r301B generated including single (C1002G, T1122C, and Macc), double (C1002G/T1122C, C1002G/Macc, and T1122C/Macc), and triple C1002G/T1122C/Macc) mutations showing all viruses only produce unspliced and gC104, not gC145. All RT-PCR reactions were electrophoresized through a 2% agarose gel.

**Table 1 viruses-16-00782-t001:** Primers used for cloning, sequencing, generation of shuttle vectors, and RT-PCR analysis.

Virus ^1^	Gene ^2^	Direction	Sequence (5′→3′) ^3^	Purpose ^4^
HVT	*UL47*	Forward	CGTGGCAAGTCTGCGGTTGG	Sequencing
		Reverse	CAACCACATTGCTTCCGC	
HVT	*UL44*	Forward	CGTAAGCTTTGTGTTTTATTGAGCGGTCG	Cloning
		Reverse	CGTTCTAGATTTGGCCGCTGCGTGATACC	
HVT	*UL44*	Forward	CGACGGGATCCCCAGGGTTCTTTCTGGACTAGTCCTACACCCCGTGGAAATAGGGATAACAGGGTAATCGATTT	Shuttle Vector
		Reverse	TTAACGGATCCGCCAGTGTTACAACCAATTAACC	
MDV	*UL44*	Forward	TATCTTCCCTCATGCTCACG	RT-PCR (full)
		Forward	CTCGGAAAGATCGCGATGGA	RT-PCR (diff)
		Reverse	CACAACTACATAACAATGAGATTATAATCG	RT-PCR (full/diff)
301B	*UL44*	Forward	CCCATGCACGCGTCACG	RT-PCR (full)
		Forward	GGGAGCCACCCTGGTATCTA	RT-PCR (diff)
		Reverse	GACGGACGAAAGTAATATGTATTTTTTCCCG	RT-PCR (full/diff)
HVT	*UL44*	Forward	CCGCAAGTATATGGTTTCCAACATGC	RT-PCR (full)
		Forward	GTGTGCCAACGACCCGCATC	RT-PCR (diff)
		Reverse	CTTAATTCCGCCCCGGTAGG	RT-PCR (full/diff)

^1^ Virus analyzed by set of primers. ^2^ Gene analyzed with the set of primers. ^3^ Restriction sites or start/stop codons are underlined. ^4^ Main purpose of primer is to amplify the full-length (full) cDNA or differentiate (diff) splice variants.

**Table 2 viruses-16-00782-t002:** Primers used for generation of recombinant HVT.

Modification ^1^	Direction	Sequence (5′→3′) ^2^
eGFP at C-term of HVT UL47	Forward	gatgccaaagatgtaatatttgaaccgcgcccttttaaa * ATG * *gtgagcaagggcgaggag*
	Reverse	aacacccctccgcgtcgggcaagagtttcggaagcacg CTA *cttgtacagctcgtccatgccg*
Removal of HVT gC (ΔgC)	Forward	tttataccatattacgcatctatcgaaacttgttcgagaaccgcaagtatTAAgattaaccatcgtat*tagggataacagggtaatcgattt*
	Reverse	ggttataacacttaataatttttatatcacatacgatggttaatcTTAatacttgcggttctcgaaca*gccagtgttacaaccaattaacc*
HVT gC into rHΔgC	Forward	tttataccatattacgcatctatcgaaact *tgt* *gttttattgagcggtcg*
	Reverse	ggttataacacttaataatttttatatcac *ctgatcagcgggtttaaacg*
HA at N-term of HVT gC	Forward	agttctgtctttacagcaaacctcttgtgccggattgccc ** tatccgtatgatgtgccggattatgcg ** *tagggataacagggtaatcgattt*
	Reverse	ttgaaagttaggatatgatgggtatcgacgttatg**cgcataatccggcacatcatacggata**gggcaatccggca*gccagtgttacaaccaattaacc*
MDV gC into rHΔgC	Forward	tttataccatattacgcatctatcgaaact *tgttcgagaatatcttccctcatgctcacg*
	Reverse	ggttataacacttaataatttttatatcac *atacgatggtcataacaatgagattataat*

^1^ Modification of the HVT genome using two-step Red recombination. ^2^ Italics indicate the template-binding region of the primers for PCR amplification with pEP-eGFP-in, pEP-KanSII, or pEP-HVTgC-in, or pEP-MDVgC-in. Red indicates unique upstream integration sequences. Green indicates unique downstream integration sequences. Start and stop codons are noted in capital letters and underlined. Nucleotides targeted for mutagenesis are noted in bold and underlined.

**Table 3 viruses-16-00782-t003:** Primers used for generation of recombinant 301B/1.

Modification ^1^	Direction	Sequence (5′→3′) ^2^
*UL44* C1002G (gC145 donor)	Forward	agcagggaaatatgatgagtactacgaacgccactgcaatcccgac**G**gtatatcatcatccccgg*tagggataacagggtaatcgattt*
	Reverse	ttgcatacccatctttgaaagccagggatatccggggatgatgatata**C**cgtcgggattgcagtgg*gccagtgttacaaccaattaacc*
*UL44* T1122C (BP)	Forward	ggaattaccatacgatggttagtacacgatgaacccaaacctaa**C**acaacttatgatact*tagggataacagggtaatcgattt*
	Reverse	cttgagggtcctgcaaagacctgtaaccacagtatcataagttgt**G**ttaggtttgggttc*gccagtgttacaaccaattaacc*
*UL44* MDV acceptor (Macc)	Forward	tagtacacgatgaacccaaacctaatacaacttat**A**atactgtggttacaggtct**C**tgc**C**ggacc**A**tc**G**a**T***tagggataacagggtaatcgattt*
	Forward *	tagtacacgatgaacccaaacctaa**C**acaacttat**A**atactgtggttacaggtct**C**tgc**C**ggacc**A**tc**G**a**T***tagggataacagggtaatcgattt*
	Reverse	aatattcggctgatgatatttctatgccg**A**t**C**ga**T**ggtcc**G**gca**G**agacctgtaaccacagtat**T***gccagtgttacaaccaattaacc*

^1^ Modification of the 301B/1 genome using two-step Red recombination. ^2^
*Italics indicate the template-binding region of the primers for PCR amplification with pEP-KanSII.* Red indicates unique upstream integration sequences. Green indicates unique downstream integration sequences. Nucleotides targeted for mutagenesis are capitalized and noted in bold and underlined. * Unique forward primer needed to maintain T1122C mutation in BP-modified rMDV.

**Table 4 viruses-16-00782-t004:** Predicted mRNA splice donor sequences for MDV, HVT, and 301B/1.

Virus	Name ^1^	Start ^2^	End ^3^	Score ^4^	Exon/Intron (5′-3′)
MDV	MD1	162	176	0.90	aactgag/gtacctca
	MD2	187	201	0.29	acagaaa/gtgtgtca
	MD3	611	625	0.25	gaaacaa/gtacttca
	MD4	751	765	0.18	tacatac/gtgtgtgt
	MD5 (gC145)	1074	1088	0.24	cccgacg/gtatatca
	MD6 (gC104)	1115	1129	0.69	aagatgg/gtatgcaa
	MD7	1166	1180	0.95	tacggtg/gttagtac
	MD8	1392	1406	0.16	acccatg/gttattac
HVT	HD1 (gC145)	1050	1064	0.15	accgacg/gtatatta
	HD2 (gC104)	1091	1105	0.69	aagatgg/gtatgcaa
	HD3	1142	1156	0.54	tgaggtg/gttagttc
	HD4	1411	1425	0.12	tgaggtg/gttagttc
301B/1	3D1	533	547	0.18	gagacaa/gtacttca
	3D2	823	837	0.99	acaccac/gtaaggac
	3D3	879	893	0.14	caccctg/gtatctac
	3D4 (gC104)	1037	1051	0.69	aagatgg/gtatgcaa
	3D5	1088	1102	0.98	tacgatg/gttagtac

^1^ Named in this report. ^2^ Donor sequence starting nucleotide in respective *UL44* gene. ^3^ Donor sequence ending nucleotide in respective *UL44* gene. ^4^ Score provided in prediction.

**Table 5 viruses-16-00782-t005:** Predicted mRNA splice acceptor sequences for MDV, HVT, and 301B/1.

Virus	Name ^1^	Start ^2^	End ^3^	Score ^4^	Intron/Exon (5′-3′)
MDV	MA1	170	210	0.70	tacctcatgcaccttccaca/gaaagtgtgtcaacaaattcg
	MA2	413	453	0.89	aatggtccaactttgttcta/gatctgatctttaacccaatt
	MA3	719	759	0.19	attttaatcggcctttaata/gataaacatatttacatacgt
	MA4	782	822	0.20	tggatgtactggcccctcca/gtcctcagcggagaaaattac
	MA5	822	862	0.11	caaggcatcttgtatcgtta/gacacttttatccccctggat
	MA6	1097	1137	0.23	gtttatccctggcttttaaa/gatgggtatgcaatatgtact
	MA7 (gC104/145)	1205	1245	0.73	cttataatactgtggttaca/ggtctctgccggaccatcgat
	MA8	1233	1273	0.42	ccggaccatcgatcgccata/gaaatctcctcagccgcattc
	MA9	1301	1341	0.31	aatatacgtgcagactcata/ggctaccccttcgatgaagat
	MA10	1421	1461	0.48	tgggattggctgtaatttta/gggatggggataatcatgact
HVT	HA1	58	98	0.96	tttctagttctgtctttaca/gcaaacctcttgtgccggatt
	HA2	155	195	0.47	ccgatggcgttcctttgtca/gaggtgcccaattcgcctacg
	HA3	323	363	0.13	tcgtattccttaacaataca/ggaagaattttgtgtgacctt
	HA4	347	387	0.33	gaattttgtgtgaccttata/gtcgaccccccttcagacgat
	HA5	362	402	0.38	ttatagtcgaccccccttca/gacgatgaatggtccaacttc
	HA6	758	798	0.69	tggatgtattggcccctcca/gttctcagcggagaaaactac
	HA7	967	1007	0.84	ggactcgaatctcctccaaa/ggtttcctgcttggtagcgtg
	HA8	983	1023	0.28	caaaggtttcctgcttggta/gcgtggaggcaaggcgatatg
	HA9 (gC104/145)	1181	1221	0.51	cttatgatactgtggttaca/ggtctctgcaggaccatcgat
	HA10	1191	1231	0.81	tgtggttacaggtctctgca/ggaccatcgatcgttatagaa
	HA11	1209	1249	0.58	caggaccatcgatcgttata/gaaatctcgccagtcggattc
	HA12	1237	1277	0.37	gccagtcggattccagtcca/ggacaactgggcgaaaacgaa
	HA13	1379	1419	0.29	taacaattacggccgttcta/ggactggccttgtttttaggt
	HA14	1397	1437	0.95	taggactggccttgttttta/ggtattggtatcattatcaca
301B/1	3A1	170	210	0.17	ttcctgcatccccgccggca/ggggagaaagaggagagccac
	3A2	227	267	0.13	cgcgtaggatgcctagtata/gtttgcgataaagaagaagtt
	3A3	292	332	0.56	cgtttcgtgtgcactcttaa/gatcgcccctccctccgacaa
	3A4	641	681	0.45	ctttcaatcggcctttacta/gataagcacgtgtacatccgc
	3A5	711	751	0.43	tctagccccccccgtcctca/gtggcgataagtacaaggctt
	3A6	744	784	0.26	caaggcttcatgcatcgtta/ggcatttttatccaccgggct
	3A7	913	953	0.36	gccatcgaccctcctcccaa/gatttcatgtctggtagcctg
	3A8	929	969	0.17	ccaagatttcatgtctggta/gcctggaagcagggaaatatg
	3A9 (gC104/145)	1127	1167	0.56	cttatgatactgtggttaca/ggtctttgcaggaccctcaag
	3A10	1137	1177	0.85	tgtggttacaggtctttgca/ggaccctcaagcggcatagaa
	3A11	1313	1353	0.10	cccccatggttctcgcgata/gcggctgttgtgggactagct

^1^ Named in this report. ^2^ Acceptor sequence starting nucleotide in respective *UL44* gene. ^3^ Acceptor sequence ending nucleotide in respective *UL44* gene. ^4^ Score provided in prediction.

## Data Availability

Data are contained within the article.

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
