# Peer review of "mRNA Splicing of UL44 and Secretion of Alphaherpesvirinae Glycoprotein C (gC) Is Conserved among the Mardiviruses"

_viruses, 2024, doi:10.3390/v16050782_

Round 1

Reviewer 1 Report

Comments and Suggestions for Authors

This work by Xu and colleagues is a comparative study on the secretion of glycoprotein gC of mardiviruses MDV, HVT and GaHV3 (strain 301B/1). The UL44 gene encoding gC is differentially spliced thus giving three proteins: full-length gC, containing a transmembrane domain (from unspliced UL44) and two secreted forms of gC that lack the transmembrane domain after splicing of UL44, gC-104 and gC-145. gC is essential for horizontal transmission of MDV and 301B/1 although the respective contribution of the different forms is unclear.

Following their previous works on secretion of gC of MDV and 301B/1, they start by investigating whether gC from HVT is also secreted. They show that it is and conclude that secretion of gC is conserved. Then, contrary to the study they published in 2021 (Vega-Rodriguez et al., Scientific Reports 2021), they look further at the mRNA level to verify that splicing of UL44 occurs in HVT and 301B/1 infected cells as it does in MDV infected cells. These analyses demonstrate that although splicing of UL44 is comparable between HVT and MDV, the splice form corresponding to gC-145 is missing for 301B/1.

This is a well-written manuscript with results that are robust and clearly described despite profusion of different viruses, constructs and spliced forms. Although the mechanism behind the regulation of UL44 splicing remains unknown, this study completes nicely the current state of knowledge on the splicing of UL44 and secretion of gC of Mardiviruses. It will provide useful information to the Herpesvirus community in general that lacks this type of studies on the regulation of glycoprotein expression and the associated inter-individual transmission.

I have a few suggestions to improve the manuscript.

1) The title says that mRNA splicing and secretion are “highly conserved”, but splicing of UL44 is only partially conserved for 301B/1 according to the authors. I would suggest either to delete “highly” or to rephrase the title.

2) The authors conclude that the gC-145 form is not detectable by RT-PCR from 301B/1 infected cells (Figure 5). The expected size of the gC-145 band is not specified. On the basis of what is observed with MDV and HVT, I assume that it should be roughly 40bp less than gC-104 and the two bands should therefore be quite close together, with gC-145 being the least abundant. I am therefore wondering whether this band could be hidden by the very large gC-104 band visible on all RT-PCR showed in Figure 5. A lower exposure of those gels could provide an answer. Are those TBE agarose gels (not specified in M&M)?

3) MDV ICP27 was shown to inhibit the splicing of gC-145 in cells transfected with plasmids encoding ICP27 and UL44 (Chuard et al., JVI 2020). In line with the authors’ discussion about the role of ICP27 in regulating UL44 splicing, an assay the authors could easily and rapidly provide would be to transfect cells with a plasmid encoding MDV, HVT or 301B/1 UL44 with or without a plasmid encoding ICP27 from MDV, HVT or 301B/1 (with testing all combinations). First, this could show that gC-145 is visible in cells transfected with 301B/1 UL44 alone and absent in the presence of ICP27. Second, it would show whether the role of ICP27 in UL44 splicing is conserved in Mardiviruses.

Other suggestions

1) The ICTV has recently updated the convention for naming Herpesviruses. For instance, Gallid alphaherpesvirus 2 (GaHV-2) is now named Mardivirus gallidalpha2 (https://ictv.global/taxonomy/taxondetails?taxnode_id=202201419&taxon_name=Mardivirus%20gallidalpha2).

2) Line 100: pICP27 is inappropriate. ICP27 is the protein, UL54 is the gene.

3) The original reference for anti-MDV gC antibody A6 is missing (the antibody is not listed in section 2.2

4) Lines 188-189 and 189-192 are redundant. Lines 188-189 could be merged into the following ones.

5) Line 196-197: Please precise how the HA tag was inserted on UL44 (at what stage in particular).

6) Lines 234 and 238: I assume the authors mean “HA” and not “1xFlag” ?

7) Figure 2 (a) : it is not easy to figure out what are the different viruses used here without consulting the Vega-Rodriguez paper.

8) Figure 2: I am surprised by the amount of secreted gC the authors can detect in the cell media, it looks like more than BSA! Did the authors perform some sort of concentration before?

9) Figure 3a is not convincing. I understand that the expected difference is tiny for PCR products of this size, but i would suggest the authors provide a better resolved gel (longer migration and an enlarged version of the gel).

10) Figure 3c: the labelling is confusing. First, the orientation is different from the other figures and second, it took me time to realize that samples were not put in the same order between CEC and FFE.

11) Line 322: I assume the authors mean Table 1 and not Table 2. In addition, the primers should be depicted on Figure 3b for better understanding.

12) Resolution of Figure 5a is rather low. Perhaps zooming only on the regions of interest would help improve readability.

13) Lines 396-401 and 416-423 state essentially the same thing. Discussion could be enriched on discussing the respective horizontal transmission of GaHV3 and GaHV2. Is transmission of GaHV3 as efficient as that of GaHV2 and how could that relate to the lack of gC-145 for GaHV3 knowing that this lab has previously shown that all three forms of gC are required for efficient transmission of MDV?

14) line 84: “with a portion of the UL45.”, “gene” is missing?

Author Response

Reviewer 1

This work by Xu and colleagues is a comparative study on the secretion of glycoprotein gC of mardiviruses MDV, HVT and GaHV3 (strain 301B/1). The UL44 gene encoding gC is differentially spliced thus giving three proteins: full-length gC, containing a transmembrane domain (from unspliced UL44) and two secreted forms of gC that lack the transmembrane domain after splicing of UL44, gC-104 and gC-145. gC is essential for horizontal transmission of MDV and 301B/1 although the respective contribution of the different forms is unclear.

Following their previous works on secretion of gC of MDV and 301B/1, they start by investigating whether gC from HVT is also secreted. They show that it is and conclude that secretion of gC is conserved. Then, contrary to the study they published in 2021 (Vega-Rodriguez et al., Scientific Reports 2021), they look further at the mRNA level to verify that splicing of UL44 occurs in HVT and 301B/1 infected cells as it does in MDV infected cells. These analyses demonstrate that although splicing of UL44 is comparable between HVT and MDV, the splice form corresponding to gC-145 is missing for 301B/1.

This is a well-written manuscript with results that are robust and clearly described despite profusion of different viruses, constructs and spliced forms. Although the mechanism behind the regulation of UL44 splicing remains unknown, this study completes nicely the current state of knowledge on the splicing of UL44 and secretion of gC of Mardiviruses. It will provide useful information to the Herpesvirus community in general that lacks this type of studies on the regulation of glycoprotein expression and the associated inter-individual transmission.

I have a few suggestions to improve the manuscript.

1) The title says that mRNA splicing and secretion are “highly conserved”, but splicing of UL44 is only partially conserved for 301B/1 according to the authors. I would suggest either to delete “highly” or to rephrase the title.

Response: We agree and have removed “highly” from the title (Line 3), along with the abstract (Line 29) and section 3.3 (Line 307) of revised MS.

2) The authors conclude that the gC-145 form is not detectable by RT-PCR from 301B/1 infected cells (Figure 5). The expected size of the gC-145 band is not specified. On the basis of what is observed with MDV and HVT, I assume that it should be roughly 40bp less than gC-104 and the two bands should therefore be quite close together, with gC-145 being the least abundant. I am therefore wondering whether this band could be hidden by the very large gC-104 band visible on all RT-PCR showed in Figure 5. A lower exposure of those gels could provide an answer. Are those TBE agarose gels (not specified in M&M)?

Response: We agree the images are a bit overexpressed, but we do not have lower exposed images available. The predicted size of 301B/1 gC145 should be 417 bp. Based on our differential PCR products, we would expect to see the following sizes (Full at 562 bp;gC104 at 458; and gC145 at 417).  Using this technique, a 40 bp difference would be clear, as is seen with MDV and HVT in figure 3c, as well as Fig. 5c with MDV gC.

These are TAE gels now noted on lines 150-151. While going over the manuscript, we failed to note the differential PCRs were run on 2% agarose gels and have included this information in the revised manuscript (lines 150, 325, and 377).

3) MDV ICP27 was shown to inhibit the splicing of gC-145 in cells transfected with plasmids encoding ICP27 and UL44 (Chuard et al., JVI 2020). In line with the authors’ discussion about the role of ICP27 in regulating UL44 splicing, an assay the authors could easily and rapidly provide would be to transfect cells with a plasmid encoding MDV, HVT or 301B/1 UL44 with or without a plasmid encoding ICP27 from MDV, HVT or 301B/1 (with testing all combinations). First, this could show that gC-145 is visible in cells transfected with 301B/1 UL44 alone and absent in the presence of ICP27. Second, it would show whether the role of ICP27 in UL44 splicing is conserved in Mardiviruses.

Response: Thank you for this suggestion. Although the proposed experiments is relatively straightforward in theory, it would be beyond the scope of this current report as it would entail cloning multiple genes (pICP27 and pUL47) from multiple viruses and performing numerous co-transfection assays that we feel would be more appropriate to address in a more detailed and expanded project addressing this question directly. No changes have been made to the manuscript.

Other suggestions

1) The ICTV has recently updated the convention for naming Herpesviruses. For instance, Gallid alphaherpesvirus 2 (GaHV-2) is now named Mardivirus gallidalpha2 (https://ictv.global/taxonomy/taxondetails?taxnode_id=202201419&taxon_name=Mardivirus%20gallidalpha2).

Response: Thank you for bringing this up. It is true that the species name of all herpesviruses was recently updated. However, the names of the viruses have not changed and therefore, using the species name is not required. I recently contacted Dr. Andrew Davison, the former Chair of the Herpesvirales Study Group for the ICTV, who was the lead in renaming the species, for clarification of the change in names. He said, “A virus is a physical thing, and a species is a conceptual or imaginary thing to which we assign a virus (or viruses). The ICTV has authority only over species names. Virus names are determined by the field, not be any statutory body and definitely not by the ICTV. So, many herpesviruses have original names (e.g. Marek's disease virus) and systematic names (e.g. gallid alphaherpesvirus 2, as well as its precursor gallid herpesvirus 2). I could say more, but the upshot of this is that is WRONG to call viruses (when considered as physical things) by their species names. So, in order to give a nod to taxonomy in a paper on a physical virus (which is almost always what is in view), the way to start an Introduction would be (for example) to say: Gallid alphaherpesvirus 2 (or Marek's disease virus; species Mardivirus gallidalpha2 in family Orthoherpesviridae) is the cause of a disease of chickens…In contrast, it would be WRONG to say something like: ‘Species Mardivirus gallidalpha2 in family Orthoherpesviridae (gallid alphaherpesvirus 2 or Marek's disease virus) is the cause of a disease of chickens...”

Unfortunately, there is a misunderstanding in the field of naming of viruses in the manuscript. Based on his description, we are correct in our naming of the viruses used in this study, but we should not italicize their names. Therefore, we have corrected the names according to Dr. Davison's suggestions. Also, abbreviations should include the subfamily initial, so GaHV2 should be GaAHV2 and MeHV1 should be MeAHV1 based on the most recent ICTV naming and abbreviations.

2) Line 100: pICP27 is inappropriate. ICP27 is the protein, UL54 is the gene.

Response: Thank you for noticing this.  We typically put a “p” to indicate we are talking about the protein and not the gene. We have removed “p” when discussing the protein and italicized the gene when the gene and protein have the same name (i.e., UL47 vs UL47).

3) The original reference for anti-MDV gC antibody A6 is missing (the antibody is not listed in section 2.2

Response: Thank you for noticing this. The original description of A6 was in Tischer and is cited there. The following is included on lines 128-129 of the revised manuscript, “for MDV gC”.

4) Lines 188-189 and 189-192 are redundant. Lines 188-189 could be merged into the following ones.

Response: Thank you for noticing that. The two sentences were combined to remove the redundancy (lines 191-194).

5) Line 196-197: Please precise how the HA tag was inserted on UL44 (at what stage in particular). : I assume the authors mean “HA” and not “1xFlag”

Response: Thank you for noticing this. We have included, “…using two-step Red recombination using primers shown in Table 2” (Lines 199-200). We have also changed the 1×Flag to HA as the reviewer noted on lines 237 and 241.

7) Figure 2 (a) : it is not easy to figure out what are the different viruses used here without consulting the Vega-Rodriguez paper.

Response: We have included further descriptions in Fig. 2 legend to better describe the viruses (Lines 298-300).

8) Figure 2: I am surprised by the amount of secreted gC the authors can detect in the cell media, it looks like more than BSA! Did the authors perform some sort of concentration before?

Response: No, there was no concentration, with say concanavalin-A, prior to western blotting. It is estimated >90% of gC protein is secreted (Ref 17). Also to note, at the time of collection of media, CECs are maintained in only 0.2% FBS so that may explain the low amount of BSA staining as well.

9) Figure 3a is not convincing. I understand that the expected difference is tiny for PCR products of this size, but i would suggest the authors provide a better resolved gel (longer migration and an enlarged version of the gel).

Response: We agree that using the RTPCR primers of the full length respective UL44 transcripts is not overly convincing but was consistent with our earlier reports on MDV UL44 splicing (Ref 18). Hence, we sought a better approach to resolve the small differences in the different transcripts shown in Fig. 3c.  We did notice we neglected to mention the NS and S labels in Fig. 3a and this is included on lines 321-322.

10) Figure 3c: the labelling is confusing. First, the orientation is different from the other figures and second, it took me time to realize that samples were not put in the same order between CEC and FFE.

Response: We agree this gel was loaded in an odd way, but we wanted to present the different viruses and tissues on the same gel. No changes have been made.

11) Line 322: I assume the authors mean Table 1 and not Table 2. In addition, the primers should be depicted on Figure 3b for better understanding.

Response: You are correct. Thank you for noticing this. We have corrected the table number. Also, we have included the following sentence to the legend (Lines 325-326), “Primer locations are noted for each gene.”

12) Resolution of Figure 5a is rather low. Perhaps zooming only on the regions of interest would help improve readability.

Response: We have updated this figure to zoom into the regions of interest as requested.

13) Lines 396-401 and 416-423 state essentially the same thing. Discussion could be enriched on discussing the respective horizontal transmission of GaHV3 and GaHV2. Is transmission of GaHV3 as efficient as that of GaHV2 and how could that relate to the lack of gC-145 for GaHV3 knowing that this lab has previously shown that all three forms of gC are required for efficient transmission of MDV?

Response: This is an interesting point and one we had not thought of before.  In our experimental model, we typically see 80-100% transmission of MDV and GaAHV3 in chickens over the course of 8 weeks. Thus, it does not appear there is a difference between the two viruses. We don’t feel it is warranted to discuss without additional data that may suggest differences in transmission.

14) line 84: “with a portion of the UL45.”, “gene” is missing?

Response: Thank you for noticing this.  We have removed “the” since UL45 is the gene.

Reviewer 2 Report

Comments and Suggestions for Authors

Xu et al., studies the splicing of gC (UL44) of two Mardiviruses, GaAHV-3, and HVT. They found, that splicing of gC in these two related viruses is very similar to splicing of this gene in Marek disease virus. They suppose that this enzimatic process (conserved in Mardiviruses) might be useful in further studies to produce more effective Marek disease vaccines.

Tables, Figures, description of the applied methods are correct. The whole text is not easy to follow, the authors should give some explanation why do they think this finding would be important in Marek vaccine development.   

A final sentence would be necessary to the end of the introduction chapter to summarize in brief the main goal of this work. After the introduction it is not clear.

In the discussion they should spent some sentences on describing why and how this achivement would serve Marek vaccine development.

The whole text should have been more concise, except the Discussion chapter.

Author Response

Review 2

Xu et al., studies the splicing of gC (UL44) of two Mardiviruses, GaAHV-3, and HVT. They found, that splicing of gC in these two related viruses is very similar to splicing of this gene in Marek disease virus. They suppose that this enzimatic process (conserved in Mardiviruses) might be useful in further studies to produce more effective Marek disease vaccines.

Tables, Figures, description of the applied methods are correct. The whole text is not easy to follow, the authors should give some explanation why do they think this finding would be important in Marek vaccine development.   

Response: We have included the following in the abstract, “to block transmission
on lines 30-31.

A final sentence would be necessary to the end of the introduction chapter to summarize in brief the main goal of this work. After the introduction it is not clear.

Response: The last paragraph summarizes the work of this report (lines 89-105).

In the discussion they should spent some sentences on describing why and how this achivement would serve Marek vaccine development.

Response: We have included a final sentence in the Conclusion, “Understanding the role these individual gC proteins play in transmission will help in generating MD vaccines that block horizontal transmission of MDV in the field” on lines 418-419.

Reviewer 3 Report

Comments and Suggestions for Authors

In this manuscript, the authors sought to investigate whether splicing of the gC mRNA in avian alphaherpesviruses.  Thoroughly via traditional cloning assays, the authors identified and exchanged splice donor/acceptor sites in GaHV3 gC mRNA and showed the resulting product was secreted.  Interestingly, splice sites utilzed in MDV did not result in splicing in GaHV3 infection, the authors hypothesize is due to the action of the virus' ICP27 homologues.  While coexpression of these homolgoues in the splicing assays would enrich the manuscript, it may also be worth considering and mentioning in the text if there could be intronic sequences that affect the proximal splicing.

In the Methods, make the 3 and 2 in CO3 and CO2 subscript

Author Response

Review 3

In this manuscript, the authors sought to investigate whether splicing of the gC mRNA in avian alphaherpesviruses.  Thoroughly via traditional cloning assays, the authors identified and exchanged splice donor/acceptor sites in GaHV3 gC mRNA and showed the resulting product was secreted.  Interestingly, splice sites utilzed in MDV did not result in splicing in GaHV3 infection, the authors hypothesize is due to the action of the virus' ICP27 homologues.  While coexpression of these homolgoues in the splicing assays would enrich the manuscript, it may also be worth considering and mentioning in the text if there could be intronic sequences that affect the proximal splicing.

Response: Thank you for this comment. We do discuss the potential for intron-responsive elements that affect splicing on lines 406-412.

In the Methods, make the 3 and 2 in CO3 and CO2 subscript

Response: Thank you for noticing this. They have been corrected.